# Immunotherapy of Clear-Cell Renal-Cell Carcinoma

**DOI:** 10.3390/cancers16112092

**Published:** 2024-05-31

**Authors:** Sophie Grigolo, Luis Filgueira

**Affiliations:** Anatomy, University of Fribourg, 1700 Fribourg, Switzerland; sophie.grigolo@unifr.ch

**Keywords:** clear-cell renal-cell carcinoma, immune checkpoint proteins, interleukin 2, immunotherapy combinations

## Abstract

**Simple Summary:**

Immunotherapeutic synergy represents the cornerstone of the treatment intended for patients with clear-cell renal-cell carcinoma. In most cases, this type of cancer manifests its first symptoms at an advanced stage, thus compromising the use of local therapies such as surgery. This paper aims to describe immune checkpoint proteins, known to suppress an anti-cancer immune response, as well as their inhibitors. Interleukin-2, a cytokine used as a therapeutic option and implicated in the activation of the immune cells, is also discussed. This review further emphasizes the role of the immunotherapeutic combination, not only as an alternative to current treatments devised to counteract the progression and metastasis of ccRCC but also as a valid solution aimed at alleviating the limitations resulting from immune-based monotherapies.

**Abstract:**

Clear-cell Renal-Cell Carcinoma (ccRCC) is the most common type of renal-cell carcinoma (RCC). In many cases, RCC patients manifest the first symptoms during the advanced stage of the disease. For this reason, immunotherapy appears to be one of the dominant treatments to achieve a resolution. In this review, we focus on the presentation of the main immune checkpoint proteins that act as negative regulators of immune responses, such as PD-1, CTLA-4, LAG-3, TIGIT, and TIM-3, and their respective inhibitors. Interleukin-2, another potential component of the treatment of ccRCC patients, has also been covered. The synergy between several immunotherapies is one of the main aspects that unites the conclusions of research in recent years. To date, the combination of several immunotherapies enhances the efficacy of a monotherapy, which often manifests important limitations. Immunotherapy aimed at restoring the anti-cancer immune response in ccRCC, involved in the recognition and elimination of cancer cells, may also be a valid solution for many other types of immunogenic tumors that are diagnosed in the final stages.

## 1. Introduction

Renal-Cell Carcinoma (RCC) accounts for about 3% of all cancers and represents the most common type of kidney cancer. Its incidence peaks at between age 55–84, with a prevalence of malignancy among men [1]. A higher incidence in Europe and North America compared to Asia and Africa has been attributed to the higher availability of diagnostic imaging in Western countries [2]. The main risk factors for RCC are cigarette smoking, obesity, hypertension, Von Hippel–Lindau disease, acquired cystic disease, and inherited predisposition [3].

RCC includes several subtypes, with clear-cell RCC (ccRCC) being the most common type and representing 80% of all renal cell cancers. Second in frequency is papillary RCC, followed by chromophobe, medullary, and collecting duct RCC. These subgroups of RCCs differ from each other in genetics, biology, and behavior. ccRCC presents the highest risk of developing metastases, thus manifesting the worst prognosis among the various RCCs. Both ccRCC and papillary RCC arise from the epithelium of the proximal tubule; chromophobe and collecting duct RCC arise from the distal tubular system. Medullary RCC originates from the renal papillae or the calyceal epithelium. Medullary and collecting carcinomas tend to affect younger patients [4].

Currently, many different types of treatments have been established for the inhibition of tumor growth and RCC metastasis, including nephrectomy, immunotherapy, target and radiation therapy, and thermal- and cryo-ablation [5].

In recent years, research has largely focused on the development of new therapies targeting the regulation of the immune response. The importance of immunotherapy in the treatment of RCC is justified by the statistics, showing that in most cases, this carcinoma is silent during its early stages of development and that the first symptoms, for example, hematuria and lumbar pain, generally denote advanced disease, often accompanied by the presence of metastases [1].

Since RCC is often detected at an advanced stage, therapies such as nephrectomy cannot be applied in many patients because the RCC is no longer encapsulated in the Gerota’s fascia but shows metastasis in the lymphatic nodes and/or growth into the renal vein. However, it is essential to note that, unfortunately, the appearance of metastasis can occur in more than one-third of cases, even following surgical treatment of the primary tumor [4].

However, immunotherapy could represent a dominant treatment in combination, if possible, with surgery to obtain a good and durable recovery. Among the best-known and most widely applied immunotherapies are immune checkpoint inhibitors (ICIs). These drugs are implicated in the suppression of specific transmembrane proteins used by cancer cells as protective strategies to immobilize the body’s defenses involved in fighting tumor growth and the subsequent development of metastases [6].

Furthermore, the application of cytokines such as interleukin-2 (IL-2) has been considered an additional and important immunotherapy directed at patients with RCC and beyond. In fact, it has been demonstrated that a high level of IL-2 can induce the regression of ccRCC [7]. In the field of cancer therapy, IL-2 has been used for the activation of lymphokine-activated killer cells (LAK). Indeed, stimulation of NK and T cells with IL-2 for several days gives rise to LAK cells, which are activated peripheral blood mononuclear cells (PBMCs) [8,9,10]. LAK cells specifically target a broad variety of tumors, and they do not show activity against normal cells [11]. Unfortunately, IL-2 immunotherapy is accompanied by substantial side effects such as acute episodes of capillary leak syndrome and reduction of neutrophil function, which are also triggered by the induction and high cytotoxicity of LAK cells [7].

However, even the administration of specific ICIs can manifest side effects. In fact, in some cases, ICIs present negative responses due to the development of strong resistance by the tumor cells. However, in other cases, ICI therapy induces satisfactory results, but only for a short period of time [12]. An additional and important limitation that has hindered the routine application of immunotherapy is the heterogeneity of renal and other tumors, which explains the failure of some drugs that otherwise would be specific to certain individual tumor cell populations [13]. Based on the mentioned negative aspects of immunotherapies, recent studies have demonstrated that the synergy between different immuno- and non-immunotherapies could represent an important resolution to these issues [6,14].

In this review, we aim to more deeply expolore some of the aspects cited above. We will particularly focus on the presentation of different immune checkpoint proteins expressed in ccRCC that are involved in the negative regulation of the immune response, including Programmed cell Death Protein 1 (PD-1), Cytotoxic T-Lymphocyte-Associated protein 4 (CTLA-4), Lymphocyte-Activation Gene 3 (LAG-3), T-cell Immunoglobulin and Mucin domain 3 (TIM-3), and T cell immunoreceptor with Immunoglobulin and Immunoreceptor Tyrosine-based Inhibitory Motif (ITIM) domains (TIGIT). Subsequently, we will cover a section on drugs, particularly the inhibitors of these transmembrane proteins which are currently adopted or under investigation in the treatment of advanced ccRCC due to their involvement in restoring physiological immune-cell activity against the growth and spread of this cancer. We will then present some synergies between various immunotherapies able to increase the efficacy of ccRCC treatment, especially when tumor cells manifest resistance to single drugs. The role of IL-2 as an additional element involved in the attempt to halt the spread of this tumor will also be discussed.

## 2. Immune Checkpoints

ccRCC is characterized by a high level of intra-tumoral heterogeneity and genomic instability. In many cases, this tumor is silent during the early stages, and the appearance of the first symptoms denotes an advanced stage of the disease, so much so that it is considered one of the most lethal urological diseases once it becomes metastatic. Hence, the delayed diagnosis of ccRCC often prevents the application of therapies, such as partial or total nephrectomy, involved in removing the localized tumor mass. Consequently, immunotherapy has become a critical component in the management of this carcinoma. In particular, the introduction of ICIs has revolutionized the treatment of this and many other types of carcinomas, conferring an important increase in survival for many patients, especially those with diffuse metastases [1,4,15].

Over the last few years, research in the field of immunotherapy has made important advances, and several immune checkpoint molecules, localized mostly on exhausted T cells, have been identified. The increase in immune checkpoint expression and consequent “T-cell dysfunction” can result from the continuous antigenic stimulation of the T cells by tumor antigens, as happens in chronic diseases and some tumors [16,17].

It has been demonstrated that these molecules can be positive or negative regulators of the antitumor immune response. In fact, co-stimulatory molecules such as CD28 and CD226 can trigger the activation of T cells, stimulated through the T-cell–receptor complex and mediated by an antigen, presented on MHC molecules by an antigen-presenting cell. On the contrary, the transmembrane proteins that we will focus on, namely PD-1, CTLA-4, LAG-3, TIM-3, and TIGIT, are involved in the downregulation of a T-cellular immune response, including an anti-cancer immune response, preventing the cytotoxic T cells from killing cancer cells. These mentioned receptor/ligand molecules are examples of immune inhibitory regulators. Some other immune checkpoints oppositely modulate the activity of immune cells, depending on the binding with two different ligands, one that activates and another that inhibits activation. An example of this type of dual-action regulator is the transmembrane protein B7-H3 and the corresponding CD28 [16].

It is important to note that the expression of PD-1, CTLA-4, LAG-3, TIM-3, and TIGIT receptors does not occur only at the level of tumor-specific T cells, namely CD8^+^ cytotoxic T cells and CD4^+^ Th1 cells, but also in other cells involved in anti-tumor immunity, such as NK cells, consequently impairing functions that are involved in the recognition of malignant cells and subsequently in anti-tumor cytotoxicity with the secretion of cytokines. Important research has identified the upregulation of some specific checkpoint molecules in both T cells and NK cells [18,19].

The expression of the transmembrane proteins discussed in this review has been detected not only on the immune cells interacting with renal carcinoma but also with several other tumors, including melanoma, non-small-cell lung cancer, breast cancer, colorectal cancer, etc. [18,20,21,22,23]. This clearly demonstrates that the study of immune checkpoints and their respective inhibitors is of fundamental importance for a large portion of cancer patients who correlate immunotherapy with a strong hope of long-term survival.

In the following paragraph, we will focus on describing the individual immune checkpoints that promote the exhaustion of specific immune cells involved in cancer cell recognition (Figure 1). We begin with PD-1, one of the best-known and most studied immune regulators in the field of cancer immunotherapy. We will then cover the transmembrane proteins CTLA-4, LAG-3, TIGIT, and TIM-3.

### 2.1. Programmed Cell Death Protein-1

Programmed cell Death protein-1 (PD-1) was identified by T. Honjo in 1992 [24].

T. Honjo and J. Allison discovered the important role of PD-1 and CTLA-4 inhibition in cancer therapy. As a result, they were awarded the 2018 Nobel Prize in Physiology and Medicine [16].

PD-1, also known as CD279 or PDCD1, is a transmembrane protein encoded by the *PDCD1* gene and expressed in tumor-specific T cells (CD8^+^ cytotoxic T cells and CD4^+^ Th1 cells), NK cells, B cells, monocytes, and dendritic cells. PD-1 is composed of 288 amino acids. The structure of this immunoglobulin-like protein comprises an extracellular IgV domain linked to a transmembrane region and an intracellular tail. The latter presents two sites, an immunoreceptor tyrosine-based inhibitory motif (ITIM) and an immunoreceptor tyrosine-based switch motif (ITSM). The binding of PD-1 to its ligands, PD-L1 (CD274) or PD-L2 (CD273), leads to the phosphorylation of these two motifs. This induces the recruitment of tyrosine phosphatases (SHP-1 and SHP-2), thereby counteracting TCR signaling potential. In fact, SHP-1 and SHP-2 recruitment results in the inhibition of the PI3K-Akt and PLCγ1-Ras/MEK/ERK pathways, which are important for T cell receptor-mediated stimulation, leading to the reduced expression of Bcl-xL (a promoter of cell survival), decreased cytokine production (IFNγ, TNFα, and IL-2), etc. SHP-1 and SHP-2 are also involved in the increased expression of BIM (a pro-apoptotic molecule). All these consequences promote the reduced activation of the cells (Figure 1). Research has identified the expression of PD-1 ligands on both immune cells (professional antigen-presenting cells for the regulation of an immune response) and tumor cells [25,26,27]. Interestingly, nuclear factor kappa-light-chain-enhancer of activated B cells (NF-kB) is one of the major transcription factors inducing PD-1 gene expression in immune cells [28]. Indeed, it has been shown that during the development of RCC, NF-kB pathways are implicated in abnormal inflammatory responses, thereby contributing to the progression of the disease. Loss of the von Hippel Lindau (VHL) tumor suppressor gene, a typical mutation found in most RCC patients that will be discussed below, seems to promote the pathological roles of NF-κB [29].

### 2.2. Antigen-4 Associated with Cytotoxic T Lymphocytes

Cytotoxic T-lymphocyte-associated protein 4 (CTLA-4), also known as CD152, is another important immune checkpoint protein involved in the suppression of immune defenses against cancer, encoded by the CTLA-4 gene. CTLA-4 was discovered in 1987, but its function was only understood in the mid-1990s [25]. The basic structure includes an extracellular IgV-like domain, a transmembrane domain, and a cytoplasmic tail [26]. The entire structure comprises 223 amino acids [30].

CTLA-4 is upregulated in tumor-specific T cells and acts as an “off” switch when it binds CD80 (B7-1) or CD86 (B7-2) on the surface of cancer cells. The binding of CTLA-4 with the ligands leads to the phosphorylation of its cytoplasmic domain, named the YVKM motif. This induces the recruitment of SHP-2 and the activation of Serine/Threonine Protein Phosphatase 2A (PP2A), which subsequently suppresses the TCR/CD28-mediated activation of the PI3K-Akt pathway. Therefore, the final result involves the limitation of TCR signaling, implicated in the activation, cytokine production, and survival of T cells (Figure 1) [16,31].

### 2.3. Lymphocyte-Activation Gene 3

Lymphocyte-Activation Gene 3 (LAG-3), also known as CD223, was discovered by Triebel and colleagues in 1990. LAG-3 is a transmembrane protein with four extracellular Ig-like domains that is encoded by the LAG3 gene. The entire molecule consists of 498 amino acids [32].

Its expression has been localized at the level of the T, NK, and B cells [33].

The independent ligands with which LAG-3 interacts are Major Histocompatibility Complex II (MHC-II), Liver and lymph node sinusoidal endothelial cell C-type lectin (LSECtin), and fibrinogen-like protein 1 (FGL1) localized on antigen-presenting cells and tumor cells. LSECtin, a cell surface lectin belonging to the C-type lectin receptor superfamily, and MHC II are ligands with the highest affinity to LAG-3 [16].

LAG-3 manifests a similar behavior to the other immune checkpoints presented in this review. Indeed, it negatively regulates proliferation, cytotoxicity function, and homeostasis of the T cells involved in the recognition and elimination of foreign cells [34]. For example, it has been demonstrated that the upregulation of LAG-3 may contribute to the exhaustion of intrahepatic and peripheral hepatitis C virus (HCV)-specific CD8^+^ T cells [35] and also of peripheral CD8^+^ T cells involved in the defense against diffuse large B-cell lymphoma (DLBCL) [36].

The attenuation of the T cell responses is mediated by the cytoplasmic tail containing three conserved motifs: an FxxL motif, a KIEELE motif, and a glutamate–proline dipeptide multiple repeats motif (EP motif), as shown in Figure 1. How exactly these motifs are implicated in LAG-3 inhibitory signaling is still under investigation. To date, there is controversial research showing their essentiality [37].

The anti-immune response attributable to LAG-3 was further confirmed in ccRCC patients through a poor prognosis predicted by the expression of this protein in association with PD-1 [38]. It has also been shown that a combination of the inhibitors of these two immune checkpoints may enhance the anti-tumor response in comparison with the single inhibition of LAG-3 [19].

### 2.4. T Cell Immunoglobulin and Mucin Domain 3

T cell Immunoglobulin and Mucin domain 3 (TIM-3), also known as Hepatitis A virus Cellular Receptor 2 (HAVCR2), is another transmembrane protein belonging to the family of immune checkpoint receptors and associated with tumor-mediated immune suppression. It was first described by V. Kuchroo and colleagues in 2002. The HAVCR2 (TIM-3) gene is involved in the encoding of this protein [39]. At the beginning, its expression was observed on CD4^+^ and CD8^+^ T cells [39]. Subsequently, it has also been observed in Th17 cells [40], T_reg_ cells [41], and innate immune cells (dendritic cells, NK cells, and monocytes) [42].

The structure is composed of a membrane distal single variable immunoglobulin domain (IgV) constituting the extracellular region, a glycosylated mucin domain of variable length located closer to the membrane transmembrane region, and an intracellular stem [43]. The entire protein structure comprises 301 amino acids [44].

TIM-3 interacts with galectin-9 (Gal-9), high mobility group box 1 protein (HMGB1), phosphatidylserine (PtdSer), and carcinoembryonic antigen-related cell adhesion molecule 1 (CEACAM-1). Binding between TIM-3 and its ligand results in NK and T cell dysfunction [21,45]. Specifically, interaction with TIM-3 ligands induces the phosphorylation of Tyr256/Tyr263, localized in the cytoplasmic region. This causes the release of human leukocyte antigen B (HLA-B)-associated transcript 3 (Bat3) and the subsequent dephosphorylation of lymphocyte-specific protein tyrosine kinase (Lck). In turn, the inactivation of Lck downregulates the ZAP70/LAT/PLCγ1/Ca2^+^ TCR signaling pathway. The final outcome of this cascade is the reduction of IFNγ production, cytotoxicity, and cell proliferation (Figure 1) [21].

Published reports indicate that TIM-3 is able to downregulate immune responses under both chronic viral and cancerous conditions and it is not only highly expressed in RCC but also in several other tumor types, including melanoma, gastric cancer, etc. [18]. A study on breast cancer discovered that tumor-specific immune cells from patients with lymph node invasion show a higher expression of TIM-3 compared to patients with no lymph node invasion. This information has not yet been confirmed regarding ccRCC [46].

Despite the critical presence of this receptor in many tumors, its single blockade is ineffective, as also demonstrated for LAG-3. Therefore, combining TIM-3 blockers with other immune checkpoint inhibitors, such as anti-PD-1, may be beneficial [19].

Expression of TIM-3 was also detected on CD8^+^ T cells in myelodysplastic syndrome (MDS), demonstrating how this ligand is not exclusively involved in the exhaustion of immune cells committed to the recognition and elimination of solid tumors. Indeed, MDS is a group of bone marrow disorders manifesting ineffective hematopoiesis and with a tendency to develop into leukemia [47].

### 2.5. T Cell Immunoreceptor with Immunoglobulin and ITIM Domains

The T cell Immunoreceptor with Immunoglobulin and ITIM domains (TIGIT), also known as WUCAM or VSTM3, is a transmembrane protein discovered by Yu et al. in 2009. It belongs to the CD28 family and is encoded by the TIGIT gene. This protein is expressed on tumor-specific T cells, NK cells, dendritic cells, and macrophages, negatively influencing their activation and function against tumor cells [18,48,49].

TIGIT is constituted by an extracellular immunoglobulin variable domain, a type 1 transmembrane domain, and a cytoplasmic tail with two inhibitory motifs, namely an ITIM motif and an immunoglobulin tail tyrosine (ITT)-like motif [23,50]. The complete structure includes 244 amino acids [51].

It is an immune checkpoint protein, which, similar to LAG-3, is capable of interacting with several independent ligands. In fact, it was initially observed that TIGIT recognizes CD155 (PVR or Necl-5). Further studies discovered that it is also able to interact with two other types of ligands, namely CD112 (PVRL2 or Nectin-2) and CD113 (PVRL3 or Nectin-3). TIGIT stimulation involves the phosphorylation of ITIM and ITT-like domains, which subsequently bind to the cytosolic adaptor Grb2. In turn, this molecule recruits SHIP-1 in order to block the PI3K and MAPK pathways, thus promoting a limitation of anti-tumor immune cell activation, a reduction in immune toxicity, and an increase in immune exhaustion (Figure 1) [18,50,52].

The expression of TIGIT is hugely variable among different cancer types. RCC is one of the tumors with the lowest expression of this protein. Variations in TIGIT expression not only differ from cancer to cancer but also due to the presence of malignant cells in the lymph nodes, which has also been observed for TIM-3. Indeed, tumor-resident NK cells from patients with lymph node metastases manifest a higher presence of TIGIT than those from patients with no lymph node metastases [18].

In summary, the expression of PD-1, CTLA-4, LAG-3, TIM-3, and TIGIT ligands in tumor cells is associated with an adverse prognosis. These transmembrane proteins are negative regulators of anti-tumor immunity. Although using different signaling pathways, the cellular mechanisms of inhibition of the various immune checkpoint proteins are similar. They mainly inhibit the degranulation of NK cells and cytokine production, the reduction of IFN-γ and granzyme synthesis by effector T cells, and the augmentation of immunosuppressive functions, including the activation of regulatory T cells (T_regs_).

In the next paragraph, we will discuss ICIs that have become an important target in cancer therapy in recent years, especially in patients with advanced or metastatic ccRCC.

## 3. Immune Checkpoint Inhibitors

Immune checkpoint inhibitors are a type of immunotherapy drug preventing checkpoint proteins from signaling through different ways, with the aim of downregulating, enhancing, or deblocking a tumor-specific immune response. Consequently, the task of these inhibitors is to prevent the “off” signal from being activated, allowing the immune cells to exhibit their tumor-specific cytotoxicity and kill cancer cells. In recent years, fruitful results have shown that this therapy may be of fundamental importance in the treatment of ccRCC, especially in the more advanced stages, i.e., for most cases [53].

The modified Glasgow Prognostic Score (mGPS), a composite biomarker using albumin and C reactive protein (CRP), has been shown to be a good predictor of outcomes in ccRCC patients treated with immunotherapy. Namely, a higher mGPS is associated with shorter Progression-Free Survival (PFS) and Overall Survival (OS) in patients with metastatic ccRCC who are treated with immune checkpoint inhibitors [54].

In the next paragraph, as summarized in Figure 2 and Table 1, we will present some immunotherapy drugs of the immune checkpoint inhibitor family. We will focus, in particular, on brief descriptions of the drugs Pembrolizumab and Nivolumab (PD-1 blockers), Ipilimumab (a CTLA-4 blocker), Relatlimab (a LAG-3 blocker), Sabatolimab (a TIM-3 blocker), and Tiragolumab (a TIGIT blocker).

### 3.1. Programmed Cell Death Protein 1 Inhibitors

PD-1 inhibitors are a group of antibodies used as immune checkpoint blockers that interfere with inhibition signals induced by the PD-1 transmembrane protein on effector immune cells. To date, the anti-PD-1 drugs Pembrolizumab (Keytruda) and Nivolumab (Opdivo) have been used for the treatment of ccRCC. Pembrolizumab is the first humanized monoclonal antibody, more precisely a humanized IgG4 (S228P) monoclonal antibody. The Food and Drug Administration (FDA, USA) approved this drug in 2021 for the treatment of different types of cancers, including RCC. Nivolumab is a human antibody belonging, like Pembrolizumab, to the IgG4a class. It was the first checkpoint inhibitor approved by the FDA in 2015 to treat patients with advanced RCC. These two drugs are given as an intravenous infusion. They can cause some side effects, including nausea and diarrhea, muscle pain, lack of energy, etc., probably by interfering with physiological immune reactions in various body systems [55,56,57].

Furthermore, with patients suffering from ccRCC, anti-PD-1 drugs manifest limited efficacy in the elimination of metastases in the central nervous system, given the modest penetrance of the blood–brain barrier [7].

In order to partially overcome the limitations that follow from this type of inhibitor, numerous studies have been focusing on the simultaneous administration of combined immunotherapies. In 2018, a combination of PD-1 inhibitors with anti-CTLA-4 (Ipilimumab) was approved by the FDA to increase the success rate in ccRCC. On the other hand, to date, the synergy between pembrolizumab and anti-VEGF pazopanib is still under investigation [53].

### 3.2. Antigen-4 Associated with Cytotoxic T Lympocytes Inhibitors

Ipilimumab (Yervoy) is an FDA-approved human IgG1κ monoclonal antibody directed against Cytotoxic T-lymphocyte-associated protein 4 (CTLA-4). It is given as an intravenous infusion. The side effects of CTLA-4 inhibitors can include fatigue, diarrhea, etc. It was also discovered that the blockade of CTLA-4 for melanoma treatment can induce grade 3 and 4 Immune-Related Adverse Events (IRAEs), including autoimmune damage in the colon, liver, and endocrine glands [25].

In order to increase the efficacy of the individual drug in patients showing resistance, a combination of Ipilimumab with Nivolumab was granted FDA approval as a first-line treatment for adults with advanced RCC [58].

The success of this drug combination is attributable to their complementary mechanisms. Anti-PD-1 is involved in the reversion of effector T cell exhaustion and consequently in the reactivation of the effector response, whereas anti-CTLA-4 plays a role in antigen-specific T cell priming by inducing their activation [14].

### 3.3. Lymphocyte-Activation Gene 3 Inhibitors

The drug used to target the LAG-3 protein is Relatlimab, a human IgG4 monoclonal antibody. Currently, it is still under clinical evaluation and is in a phase II trial for ccRCC. Recent studies have demonstrated that anti-LAG-3 monotherapy and/or combinations of anti-LAG-3 and anti-PD-1 represent a good treatment for LAG-3^+^ ccRCC patients who develop an important resistance to anti-PD-1 drugs. Relatlimab is a drug administrated intravenously. It can induce side effects, including fatigue, rash, painful joints, etc. Combinations of this drug with other immune checkpoint antibodies, such as Nivolumab, are also under investigation in order to improve the effectiveness of these therapies in the suppression of this type of RCC [38].

### 3.4. T Cell Immunoglobulin and Mucin Domain 3 Inhibitors

Sabatolimab is a first-in-class immuno-myeloid therapy that targets and inhibits the transmembrane protein TIM-3. This drug is a humanized monoclonal antibody (IgG4) and is still under clinical evaluation for the treatment of several immunogenic cancers [59,60].

### 3.5. T Cell Immunoreceptor with Immunoglobulin and ITIM Domains Inhibitors

Tiragolumab is a human IgG4κ monoclonal antibody targeting TIGIT. Currently, this drug is still under clinical evaluation (phase II) for metastatic RCC. TIGIT represents a promising target in cancer immunotherapy, particularly in combination with a PD-1 inhibitor. There is evidence that blocking TIGIT and PD-1 leads to significantly increased cell proliferation, degranulation, and cytokine production by tumor-specific T cells. Potential side effects are similar to other immune checkpoint inhibitors, namely feeling tired or weak, chills, nausea, etc. [23,61].

### 3.6. Combining Other Therapeutical Approaches with ICIs

Targeting immune checkpoint proteins with humanized monoclonal antibodies (mAbs) seems to be a good solution to allow tumor-specific immune cells to upregulate and maintain their effector function, which involves the recognition and destruction of tumor cells. Unfortunately, like any cancer therapy, ICIs also present challenges. These include, for example, the occurrence of Immune-related adverse events (IRAEs), such as the already-cited autoimmune damage involving the colon, liver, and endocrine glands resulting from CTLA-4 inhibition. It has been demonstrated, although in rare cases, that ICIs can also induce the development of renal toxicity. Consequently, this may require drug suspension, renal biopsy, and/or immunosuppressive therapy administration. For this reason, it is recommended that a urinalysis and an assessment of creatinine levels are carried out before the use of these anti-cancer immune response activators and then monthly afterwards [62]. Another important difficulty resulting from the administration of ICIs is the onset of the aforementioned development of resistance by the cancer cells. According to the occurrence and timing of the latter, it is possible to distinguish a primary resistance, in never-responder patients, from a secondary resistance which occurs after an initial period of response to treatment. Research in recent years has shown that one possible way to mitigate this limitation is to focus on the synergy between different immune checkpoint blockers, as was described above with some examples, or to combine these drugs with other therapies acting on different fronts, which can have an immune or non-immune action [14,16].

In that respect, IL-2 must be considered and will be briefly discussed in the next paragraph. IL-2 is a cytokine used in immunotherapy for the treatment of ccRCC through its ability to activate cancer cell-specific immune cells, representing another example of valuable treatment alongside the ICIs [7].

Other drugs suitable for combination with ICIs include Tyrosine Kinase Inhibitors (TKIs), which are a group of pharmacological agents not currently used in immunotherapy but which, when combined with currently existing immune drugs, can support the reactivation of an anti-cancer immune response, instilling great hope in the fight against advanced-stage ccRCC. To date, research is particularly concentrated on the study of Sitravatinib, a TKI able to target both TAM receptors (such as TYRO3 and AXL) and VEGFR. The inhibition of TAM receptors reduces macrophage polarization towards a pro-tumor M2-like phenotype and apoptotic cell clearance, a tumor-promoting process called efferocytosis. On the other hand, anti-angiogenic activity, resulting from VEGFR blocking, induces the normalization of the tumor vascular structure, leading to a variety of consequences including an increase in immune cell infiltration and hypoxia mitigation. The latter leads, in turn, to the limitation of macrophage differentiation in the M2 phenotype, supporting the anti-tumor effect obtained with the TAM receptor targeting. The inactivation of VEGFR also determines the reduction in the levels of cells presenting immune-suppressive functions, such as Myeloid-Derived Suppressor Cells (MDSCs) and T_reg_ cells [14].

T_reg_ cells in the tumor microenvironment (TME) are involved in CD8^+^ T cell dysfunction, followed by subsequent exhaustion, through IL-35 secretion, which promotes the expression of several immune checkpoints on these immune cells [63].

Overall, the action of Sitravatinib may restore an immune-sensitive TME that would improve the efficacy of the ICIs. Currently, the combination of Sitravatinib and Nivolumab, for the treatment of metastatic or advanced ccRCC, is open-labelled in a phase 2 trial [14].

Several studies have shown that the composition of the gut microbiome can improve clinical outcomes of ICI therapy through the stimulation of pro-inflammatory cytokine production [63]. Therefore, the hypothesis that fecal microbiota transplantation or the manipulation of its composition can decrease the development of ICI resistance, thereby increasing the efficacy of this class of immunotherapy, has arisen. Currently, clinical trials are ongoing to validate these possible strategies [64].

## 4. Role of the von Hippel Lindau (VHL) Tumor Suppressor Gene

Around 60% to 90% of sporadic cases of RCC present with a mutation in or loss of the von Hippel Lindau (VHL) tumor suppressor gene. It has been demonstrated that, in ccRCC, this condition can induce the stabilization of hypoxia-inducible factors-αs (HIF1α and 2α), which in turn can promote metabolic reprogramming, angiogenesis, and epithelial–mesenchymal transition (EMT) associated with the downregulation of tight junction protein (occludin, claudin-1, and E-cadherin) expression and apical–basal polarity loss. The combination of these factors subsequently contributes to the growth and spread of the most common RCC subtype [29,63,65].

It is important to note that, exclusively regarding ccRCC progression, HIF1α and HIF2α present a dichotomy effect. Indeed, HIF1α acts as a tumor suppressor, while HIF2α manifests an oncogenic potential. In this regard, in addition to the therapies already described, research has recently also focused on the development of a medicament aimed at the direct inhibition of HIF2α, named Belzutifan. This therapy is targeted towards adult patients with VHL-associated RCC, as well as those suffering from central nervous system hemangioblastomas or pancreatic neuroendocrine tumors, and has been approved by the Food and Drug Administration (FDA) [29,66].

## 5. Interleukin-2

Interleukin-2 (IL-2) is a cytokine that was FDA-approved in 1992 for the treatment of advanced RCC. It is able to activate LAK cells, which are activated PBMCs, specific to a wide variety of tumors, including renal carcinoma, as shown in Figure 3 [8,67,68].

The precise mechanism applied by LAK cells to kill tumor cells is still unknown. What has been discovered is that these phenotypically IL-2-induced immune cells are able to secrete TNFα and IFNγ [67], as well as cytotoxic molecules, namely perforin, granzyme A, and granzyme B [69]. Perforins are pore-forming proteins that aggregate to form pores in the membrane of the target cell. These pores cause osmotic lysis and damage the mitochondria; they also provide a method for granzymes to penetrate the cancer cells. Once inside, granzymes trigger programmed cell death pathways [10]. The liberation of these proteins seems to be MHC restriction-independent against malignant cells [70]. It is also assumed that the action of LAK cells is not triggered by direct interaction with tumor cells but rather by a combination of cell-to-cell-mediated killing with the release of the cytotoxic proteins, which can therefore inhibit tumor growth at a distance from the effector cells (Figure 3) [71].

The activation of LAK cells involved in the fight against tumor growth thus represents an indirect action that IL-2 manifests on malignant cells. We hypothesize, however, that IL-2 therapy may also affect these cells directly. This assumption arises following the discovery of IL-2Rβ (CD122) expression in RCC, which is a receptor also shared by IL-15 [72]. The presence of IL-2Rα (CD25) and IL-2Rγ (CD132, γc) subunits have also been demonstrated, which together with IL-2Rβ form the functional heterotrimeric IL-2Rαβγ complex involved in the possible direct action of IL-2 on RCC (own unpublished data). Nevertheless, further investigation would be needed to clarify this concept.

High doses of IL-2 are often associated with severe side effects when intravenously administered, including fever, chills, hypotension, oliguria, vomiting, acute episodes of capillary leak syndrome, significantly reduced neutrophil function, etc. [73]. For these reasons, many oncologists tend to apply this therapy only to patients who are healthy enough to tolerate the side effects and whose cancer does not respond to other types of immunotherapies. During the administration of this recombinant interleukin, the physiological functions of patients should be closely monitored [74]. Consequently, the use of IL-2, which is essential to maintain LAK activation, does not represent a mainstay of RCC treatment. In fact, the administration of IL-2 therapy is limited due to its toxicity [68].

Different possible combinations between IL-2 and other therapies have been investigated in order to decrease the numerous side effects that the administration of this interleukin entails, and also to increase its effectiveness in RCC patients. The proposed synergies include, for example, the addition of IL-2 to immune checkpoint inhibitors, such as pembrolizumab, or to adoptive cellular therapy, in particular, Tumor-Infiltrating Lymphocyte (TIL) therapy [7,73].

The validity of IL-2 therapy in increasing life expectancy cannot be associated with all tumor types. Indeed, it has been hypothesized that contrary to some RCC patients, the expression of IL-2 and the heterotrimeric IL-2Rαβγ complex is associated with cancer malignancy, as shown in breast cancer [75].

## 6. Discussion and Conclusions

PD-1, CTLA-4, LAG-3, TIGIT, and TIM-3 are immune checkpoint proteins involved in the downregulation of the immune response aiming at the recognition and elimination of tumor cells. Numerous studies have shown that blocking the signaling of these immune regulators is a valuable solution to increase survival in many patients with ccRCC or other cancers. In particular, as mentioned, the combination of several monoclonal antibodies targeting ICIs has been considered to be truly essential in an attempt to overcome resistance to monotherapy by increasing its efficacy.

Regardless of the question of the synergy that can be created between several drugs, the therapy involved in the restoration of exhausted immune cells remains, in principle, the most indicated therapeutic strategy in the treatment of patients with ccRCC. This is justified by several aspects, mainly that this immunogenic type of cancer is often diagnosed at a more advanced stage.

In addition to studying combinations of ICIs to increase their effectiveness, it is crucial to understand whether the patient is likely to respond before treatment administration. In this sense, different researchers have recently focused on identifying robust predictive biomarkers to improve response rates, promoting consequent tumor regression. For instance, it has been demonstrated that immune-inflamed tumors are more sensitive to ICI therapies compared to immune deserts. A non-invasive method such as blood analysis can also be used to predict the response to these treatments. In fact, it has been noted that the expression of immune checkpoints in the periphery [20], as well as the composition of the microbiome as previously discussed [63], can contribute to the success of ICIs.

In our opinion, the most specific and precise way to understand the impact of ICI treatment is given by the expression analysis of these cell-surface receptors, and of their ligands involved in the exhaustion of anti-tumor immune cells, by biopsies of the ccRCC. To date, this is not yet considered a basic routine.

Clearly, this possibility could also entail limitations, including the heterogeneity of the protein expression within tumors and the fact that in certain cases, biopsies cannot be performed before the treatment is administered.

## Figures and Tables

**Figure 1 cancers-16-02092-f001:**
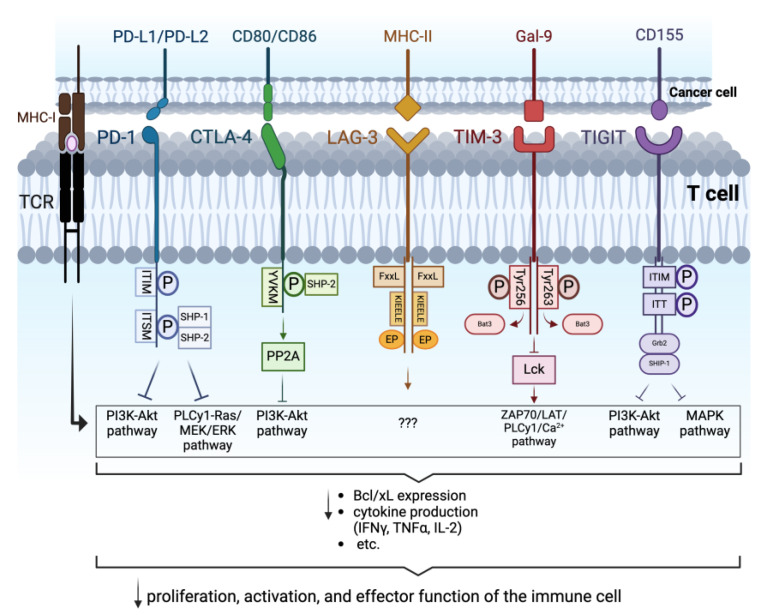
Intracellular signaling triggered by the interaction of PD-1, CTLA-4, LAG-3, TIM-3, and TIGIT with their respective ligands: the binding between PD-1 and PD-L1/PD-L2 induces phosphorylation of ITIM and ITSM, constituting the cytoplasmic domain of PD-1. SHP-1 and SHP-2 are subsequently recruited, leading to the inhibition of the PI3K-Akt and PLCy1-Ras/MEK/ERK pathways. CD80/CD86 interacting with CTLA-4 triggers the phosphorylation of the intracellular YVKM domain and the consequent recruitment of SHP-2 and activation of PP2A, which hamper the PI3K-Akt pathway. The exact mechanism triggered by the cytoplasmic tail of LAG-3, consisting of the FxxL, KIEELE, and EP domains, upon interaction with its ligand MHC-II, has not yet been clearly defined. Binding of TIM-3 to Gal-9 induces the phosphorylation of Tyr256 and Tyr263 in its intracellular tail resulting in the displacement of Bat-3 and consequently on the dephosphorylation of Lck, which leads to the downregulation of ZAP70/LAT/PLCγ1/Ca^2+^ pathway. Following the binding of TIGIT to CD155, the ITIM and ITT domains of the intracellular tail are phosphorylated. The complex binds then to Grb2 and recruits SHIP-1 which promotes the inhibition of PI3K-Akt and MAPK pathways. Overall, these signalings prevent the optimal activation and function of the immune cells involved in hindering tumor development (Arrows in the image indicate active pathways. Arrows underneath the curly brackets indicate “decrease”. “???” indicate that the pathway implicated is unknown). Created with BioRender.com.

**Figure 2 cancers-16-02092-f002:**
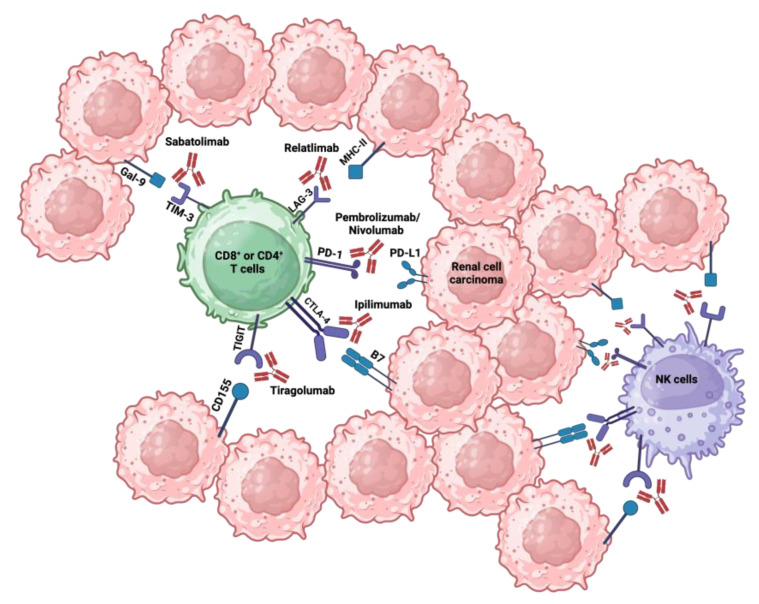
Immune checkpoint receptors associated with their respective ligands and inhibitors: immune checkpoints such as PD-1, CTLA-4, LAG-3, TIM-3, and TIGIT, expressed on T and NK cells, bind their respective ligands on RCC cells, triggering a negative signal to the immune cell response. Created with BioRender.com.

**Figure 3 cancers-16-02092-f003:**
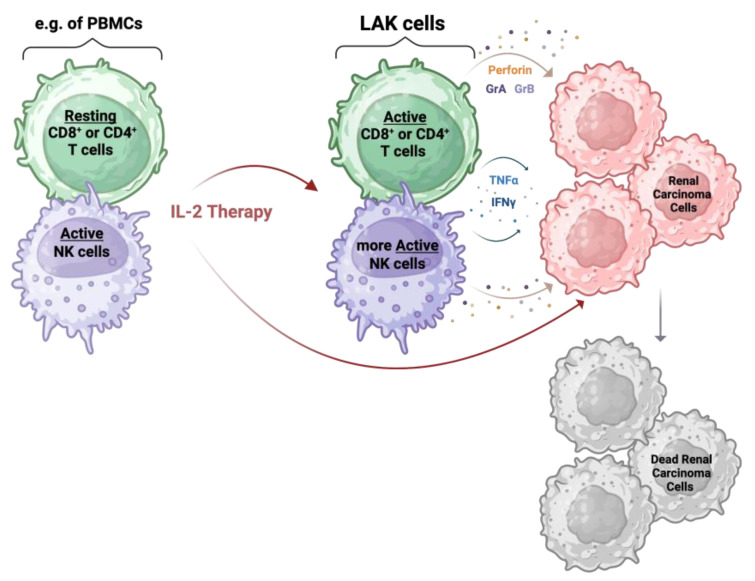
IL-2 therapy mechanism: IL-2 influences tumor growth directly and indirectly. The indirect action is induced by the IL-2 stimulation of PMBCs, especially T and NK cells, for a few days. This gives rise to LAK cells that show cytotoxic abilities against tumor cells, including renal carcinoma cells, via the release of perforin, granzyme A (GrA), granzyme B (GrB), TNFα, and IFNγ. The direct action of IL-2 is driven by the presence of an IL-2 receptor expressed on the tumor cells. Created with BioRender.com.

**Table 1 cancers-16-02092-t001:** Summary of the immune checkpoint inhibitors (ICI) drugs involved in the management of renal cell carcinoma (RCC).

ICI Drugs	Target Protein	Clinical Trial	Phase	Response Rate
Pembrolizumab	PD-1	NCT03142334 (KEYNOTE-564) (type: secondary)	Phase 3	OS 72 months
Nivolumab	PD-1	NCT01668784 (CheckMate 025) (type: secondary)	Phase 3	ORR 25.9% PFS 4.21 months
Nivolumab + Ipilimumab	PD-1 + CTLA-4	NCT02231749 (CheckMate 214) (type: primary)	Phase 3	ORR 41.6% PFS 11.56 months
Relatlimab + Nivolumab	LAG-3 + PD-1 +	NCT02996110 (FRACTION-RCC) (arm 2) (type: primary)	Phase 2	mDOR 32.57 weeks
Sabatolimab	TIM-3	NCT02608268 (type: secondary)	Phase 1–2	PFS 1.8 months OS 4.1 months
Tiragolumab + Tobemstomig + Pembrolizumab + Axitinib	TIGIT + PD-1 + LAG-3 + PD-1 + VEGFR-1	NCT05805501	Phase 2	No Study Results Posted

Abbreviations: NCT: number of clinical trial (https://clinicaltrials.gov/, accessed on 18 May 2024); OS: Overall Survival; ORR: Objective Response Rate; PFS: Progression-free Survival; mDOR: median Duration of Response.

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
