# Peer review of "Immunotherapy of Clear-Cell Renal-Cell Carcinoma"

_cancers, 2024, doi:10.3390/cancers16112092_

Round 1
Reviewer 1 Report
Comments and Suggestions for Authors
Grigolo and Filgueira have written a comprehensive review of immunotherapy in ccRCC. They cover the different check point proteins expressed in ccRCC, their mechanism of action and pathways affected, and then review the inhibitors of these proteins that are approved for or under investigation in the treatment of advanced ccRCC. They discuss combination therapies of ICI and TKIs that are being considered or have shown promise in ccRCC and finally cover IL-2 efficacy in this cancer. The review is well written, quite comprehensive and the material is presented in a logical and organized manner with several good summary illustrations. References are current for the most part.
There are a few points that should be addressed:
Pg 10, Role of VHL tumor suppressor gene, paragraph 2: in discussing Belzutifan, the authors should cite the reference: T. K. Choueiri et al., Phase III study of the hypoxia inducible factor 2α (HIF-2α) inhibitor MK-6482 versus everolimus in previously treated patients with advanced clear cell renal cell carcinoma (ccRCC). J Clin Oncol 38, TPS5094 (2020) that validates is efficacy in sporadic ccRCC.
Pg 7, Paragraph 5, Line 4: use of the word “curing” is not accurate for this reference which showed 70% objective response rate in the combination IL-2 and pembrolizumab study. Please modify that word choice.
Pg 11, Conclusion, Line 1: PD-1 is missing from the list of immune check point proteins.
Author Response
Pg 10: Recommended reference has been included in the text and in the reference list under number 65.
Pg 7: The sentence has been changed accordingly and is highlighted in the resubmitted text: ¨In fact, it has been demonstrated that a high lever of IL-2 can induce the regression of ccRCC [7].¨ (on page 2 of pdf file in the middle)
Pg 11: PD-1 has been added, and the corresponding text has been highlighted. (on page 3 and page 13 of pdf file)
Reviewer 2 Report
Comments and Suggestions for Authors
1. We need a summary table that would combine all the drugs that are used for immunotherapy of renal cell cancer, indicating the clinical trials (if any) that are/have been conducted for this drug in this cancer.
2. It is known that the main disadvantage of ICT therapy is the low response rate to treatment. For each group of inhibitors, please provide the response rate for renal cell carcinoma and how to identify patients who are more likely to respond. What is the proportion of such patients in the total number of patients with renal cell cancer?
3. I didn’t see any specific data in the review: if the drug is effective, how much does overall and disease-free survival change? What drug combinations were tested? Combinations of drugs for immunotherapy alone or, for example, in combination with β-blockers, etc.
Author Response
- Table 1 in the revised manuscript has been added with known clinical trials and their corresponding NCT numbers.
- The discussion has been accordingly adapted and this issue has been taken and discussed in the 3rd paragraph of section 6) Discussion and conclusion (highlighted).
- This issue has been included in the revised manuscript in Table 1, in the right column covering the response rate, i.e. survival information.
Reviewer 3 Report
Comments and Suggestions for Authors
The manuscript by e Grigolo and Filgueira is a review article focusing on recent advances in the field of immunotherapy of clear cell renal cell carcinoma. Although the topic of this review is timely and emerging, I would like to raise a number of questions, the answers to which I hope will improve the current manuscript.
- Sections 2.1-2.5 are devoted to specific transmembrane proteins. In my opinion, a figure with a schematic representation and thus a comparison of their structures will contribute to the understanding of further readers.
- Section 3 is dedicated to immune checkpoint inhibitors, so paragraphs 3.1-3.6 describe specific ones. However, any attempt to perform the analysis and their comparisons is lacking. The authors should present a comparison of inhibitors, pointing out their advantages and limitations. What are the most perspective?
- The conclusions are formal. However, it would be interesting to know the authors' opinion about the main directions of development of a field. What are the most perspective compounds, strategies of their use, possible combinations, etc.
- I have some concerns about the style. For example, references are given by authors at the end of paragraphs, rather than at the end of specific sentences containing specific statements that need to be supported by the specific reference. This should be corrected.
- The use of "these inhibitory receptors" in the first sentence of a new paragraph (see page 5) refers to the previous sentence, which does not exist.
Author Response
corresponding detailed figure legend has been added to the revised manuscript.
Section 3: For the improvement of the reviewer’s suggestion, Table 1 with corresponding information generated through clinical trials has been added.
Discussion: This part has been extended and hopefully adding the information the reviewer is asking for.
In text referencing: This has been adjusted, where it makes sense from our perspective.
Round 2
Reviewer 2 Report
Comments and Suggestions for Authors
I have no further comments on the manuscript.
Reviewer 3 Report
Comments and Suggestions for Authors
the manuscript has been improved